# OpenReview forum: "Convergence of Consistency Model with Multistep Sampling under General Data Assumptions"
_ICML.cc/2025/Conference — ICML 2025 poster_

### Official Review · Reviewer_VSKv · 2025-03-12

**Overall Recommendation:** 2

**Summary:**

This paper analyzes the convergence of consistency models under approximate self-consistency. With mild data assumptions, it proves sample closeness to the target distribution in Wasserstein or total variation distance. The study applies to various forward processes and highlights the benefits of multistep sampling through case studies.

## Update after rebuttal
I initially had no significant issues with the paper, but after reading Reviewer eZRV's comments, I believe the novelty of the work is somewhat limited, and my original score of 3 may have been overly generous. To ensure fairness in the review process, I have adjusted my score to 2.

**Claims And Evidence:**

Yes.

**Essential References Not Discussed:**

No.

**Experimental Designs Or Analyses:**

Yes. I checked the simulation in appendix G.

**Methods And Evaluation Criteria:**

Yes.

**Other Comments Or Suggestions:**

NO.

**Other Strengths And Weaknesses:**

Strengths:

1. The paper provides theoretical guarantees for multi-step generation in consistency models, focusing on Wasserstein distance and total variation distance. The results seem solid.

2. Compared to previous work, the paper conducts its analysis under more general assumptions, including the removal of the Lipschitz assumption for the consistency function.

3. The results are not limited to the form of SDEs and provide a more detailed analysis under two common types of SDEs.

Weaknesses:

1. The assumptions in the paper still differ from practical scenarios. For instance, the paper assumes that time discretization is uniform, which does not align with practical applications. In fact,  carefully designed  time discretization strageties are crucial for successful consistency training.

2. The analysis and experiments in the paper lack connection with real-world data, which limits the further extension of the theoretical results.

**Questions For Authors:**

Can the experiments in Appendix G be replicated on real-world datasets, such as CIFAR-10? This would significantly enhance the persuasiveness of the theoretical results.

**Relation To Broader Scientific Literature:**

The article presents the theoretical performance of consistency models with multi-step sampling under more general assumptions. The findings of the article can contribute to a deeper theoretical understanding of consistency models and can assist in the design of time steps for multi-step sampling in consistency models.

**Theoretical Claims:**

Yes. I’ve checked the correctness of proofs for theorems 2 and 3.

---

> ### Author Rebuttal · Authors · 2025-04-01
>
> Thank you for your feedback. We address your points in detail below:
> 1. **non-uniform discretization:** in this paper, we adopt a uniform discretization for clarity and ease of presentation. However, our results can be extended to the non-uniform discretization setting as well. Suppose $\tau_{0:M}$ is an arbitrary discretization of the interval $[0,T]$. In this scenario, it is reasonable to assume that the consistency loss scales with the length of the discretization interval:
>   $$
>   E_{x_{\tau_i}\sim P_{\tau_i}}\left[||\hat f(x_{\tau_i},\tau_i)-\hat f(\varphi(\tau_{i+1};x_{\tau_i},\tau_i),\tau_{i+1})||\_2^2\right]
>   \le (\tau_{i+1}-\tau_{i})^2\epsilon^2.
>   $$
> Using the same argument as in the proof of Lemma 2, we can derive:
>   $$
>   \sqrt{E_{x_{\tau_i}\sim P_{\tau_i}}\left[||\hat f(x_{\tau_i},\tau_i)- f^{\star}(x_{\tau_i},\tau_{i})||\_2^2\right]}
>   \le \sum_{s=0}^{i-1}\sqrt{
>     E_{x_{\tau_s}\sim P_{\tau_s}}\left[||\hat f(x_{\tau_s},\tau_s)-\hat f(\varphi(\tau_{s+1};x_{\tau_s},\tau_s),\tau_{s+1})||\_2^2\right]
>   },
>   $$
>   which is upper bounded by:
>   $
>   \sum_{s=0}^{i-1} (\tau_{s+1}-\tau_s)\epsilon = \tau_i \epsilon
>   $. The rest of the proof remains unchanged.
> 2. **regarding experiments on real-world data:** in designing the experiments, we aimed to achieve two goals:
>    - **evaluate the tightness of our upper bound:** this requires access to the true data distribution, which is not available for real-world datasets, making such evaluation infeasible in those settings;
>    - **demonstrate diminishing performance with multiple sampling steps:** we already observe this phenomenon in real-world datasets, as shown in Table 1 and 2 of Luo et al. (2023), where the performance of the consistency model deteriorates with more sampling steps.
> - Given these considerations, we prioritized simulation-based experiments over real-world datasets.

---

> > ### Comment · Reviewer_VSKv · 2025-04-02
> >
> > I would like to thank the authors for addressing my questions and concerns. I will maintain my evaluation.

---

### Official Review · Reviewer_kTQM · 2025-03-13

**Overall Recommendation:** 3

**Summary:**

The paper studies the convergence of consistency models with assumptions on the consistency property. It further assumes that the target data distribution has bounded support. In this case, it shows the convergence result in Wasserstein distance and total variation distance. The theoretical results indicate the benefit of multistep sampling with consistency models.

**Claims And Evidence:**

I think the presentation of this paper should be further improved. The main results are divided into two parts. In the first part (Section 3), the paper proves a error bound of the Wasserstein (TV) distance in the general form of time schedules. The meaning of the theorem, which contains very complex formulas, is not very clear to me. For explanation, the paper claims there exists a trade-off about the sampling steps. However, the result is still very hard to understand, because the parameter $\alpha_t$, $\sigma_t$ and $t_j$ has some intrinsic constraints. I find it too easy to draw the conclusion that there is a tradeoff. I’m also unclear about what this tradeoff definitely means in practice.

With these uncertainties in mind, I turned to the second part, which addresses the case studies. However, reading this section only deepened my confusion. Take the first case (VP) as an example: the paper presents a result concerning the Wasserstein distance between $\hat P_0^1$ ($\hat P_0^2$) and the data distribution. It argues that the leading term from the second sampling step is strictly reduced. While correct, this reduction merely adds at most another constant term—why, then, is this significant? Moreover, the paper considers the case $\epsilon \approx \Delta \tau$. In this case, the total right-hand side is $O(1)$. Yet, considering $R$ as a constant, any Wasserstein distance trivially has an upper bound of $O(1)$. Thus, I find myself questioning the significance and relevance of the stated result in this specific scenario.

Finally, in Lines 226-231, the paper discusses the influence of $\Delta \tau$. I don’t quite get what is “more intermediate steps” even after reading Lemma 2. I also doubt the claim that “smaller $\Delta \tau$ allows a smaller $t_N$”, and “may decrease $\epsilon$”. What’s the support of this argument? $\tau_i$ for training is independent from $t_i$ for sampling, is that right?

**Essential References Not Discussed:**

N/A

**Experimental Designs Or Analyses:**

N/A

**Methods And Evaluation Criteria:**

N/A

**Other Comments Or Suggestions:**

See above

**Other Strengths And Weaknesses:**

See above

**Questions For Authors:**

N/A

**Relation To Broader Scientific Literature:**

N/A

**Theoretical Claims:**

I don’t find flaws in the theoretical proof. My concerns mainly lie in the meaning of the results, as written in the “Claims and evidence” part.

---

> ### Author Rebuttal · Authors · 2025-04-01
>
> Thank you for your feedback. We address your points in detail below:
> 1. **Interpretation of Theorem 2:** trade-off means increasing the number of sampling steps does not necessarily lead to improved performance due to the influence of term (ii). This is in contrast to standard diffusion models, where, according to Theorem 2 of Chen et al. 2023b, the discretization error diminishes as the number of sampling steps increases, guaranteeing performance improvement with more steps.\
> For Case Study 2, we now show that a **uniform sampling** schedule $t_j = (N-j+1) \Delta\tau$ with $N = \frac{t_1}{\Delta\tau}$ (**more** sampling steps) can yield a **larger** upper bound than the **halving schedule** defined in equation (13). Ignoring absolute constants, the upper bound in equation (12) becomes:
>   $$
>   R\left(\frac{R^2}{t_1^2} + \frac{t_1}{\Delta\tau}\frac{\epsilon_{\text{cm}}^2}{\Delta\tau^2}\right)^{1/4} + \epsilon_{\text{cm}},
>   $$
>   which is minimized when $t_1 = \frac{2^{1/3}R^{2/3}\Delta\tau}{\epsilon^{2/3}}$, yielding a minimum value of:
>   $$
>   R^{7/6}\frac{\epsilon_{\text{cm}}^{1/3}}{\sqrt{\Delta\tau}} + \epsilon_{\text{cm}}.
>   $$
>   In contrast, the halving schedule achieves an upper bound of $\tilde O\left(R\sqrt{\frac{\epsilon_{\text{cm}}}{\Delta\tau}}\right)$, which is **strictly smaller** than that of the uniform schedule.\
>   Practical evidence further supports this trade-off:
>     - Our simulation results (Appendix G) show that both baseline methods experience degraded performance in the final sampling steps.
>     - In Table 1 and 2 of (Luo et al. 2023), LCM with 4-step sampling achieves better FID scores than with 8 steps.
> - Both theoretical analysis and empirical observations highlight the importance of **stratigically designing** the sampling schedule. Thus, we believe it is fair to emphasize the trade-off in choosing the number of sampling steps.
>
> 2.
>   - **clarification on the result in Case Study 1:** Corollary 1 provides a **universial upper bound** on the $W_2$ distance for the VP process, without imposing constraints on $R$, $\Delta\tau$, and $\epsilon_{\text{cm}}$. Naturally, when the consistency model $\hat f$ is poorly estimated, the generated sample quality deteriorates. We believe the case you mentioned belongs to this category. If $\frac{\epsilon_{\text{cm}}}{\Delta\tau}\approx R$, it indicates that $\hat f$ is a poor approximation. Even when using the true marginal $P_T$ as input, Lemma 2 shows that $\hat f$ incurs an error of $O(R)$. In this scenario, the resulting upper bound is inevitably loose and uninformative.
>   - **the significance of constant reduction in Case Study 1:** while many theoretical results focus on asymptotic rates and ignore constants, **constant reductions** can have significant practical implications. For example, latent diffusion models (Rombach et al.) demonstrate improved performance (e.g., lower FID scores) even with fractional gains, enabling high-resolution image synthesis in practice.
>   - **the rate improvement in Case Study 2:** our analysis in Case Study 2 shows a **clear rate improvement** for the VE process when sampling with multiple steps. If we use only a single step ($N=1$), equation (12) simplifies to:
>     $$
>     \frac{R\sqrt{R}}{\sqrt{t_1}} + t_1 \frac{\epsilon_{\text{cm}}}{\Delta\tau},
>     $$
>     with the minimum value being $R\left( \frac{\epsilon_{\text{cm}}}{\Delta\tau} \right)^{1/3}$.
>     In contrast, using the specialized schedule from equation (13), we obtain a faster rate of $\tilde O(R\sqrt{\frac{\epsilon_{\text{cm}}}{\Delta\tau}})$.
> 3. **Clarification on $\Delta\tau$:** fix a $T$, decreasing $\Delta\tau$ results in a finer discretization $\mathcal{T}$ (line 147), thereby increasing the number of discretization points $M$. This adds more terms to the error decomposition (line 419-423) when evaluated at $\tau_M = T$.
>    - **Training steps vs sampling steps:** for simplicity, our current formulation assumes sampling steps are chosen from the training steps. Under this assumption, a smaller $\Delta\tau$ allows a smaller $t_N$. However, our theoretical framework remains valid even when sampling steps are chosen arbitrarily. In that case, we need to refine Lemma 2 to upper bound $||\hat f(\cdot,t)-f(\cdot,t)||_2^2$ for all $t$. Because self-consistency is only enforced at the $\tau_i$'s during training, $\hat f(x,t)$ and $f(x,t)$ need to be Lipschitz in $t$.
>    - **Effect of $\Delta\tau$ on $\epsilon$:** using a smaller $\Delta\tau$ may reduce the consistency error $\epsilon$ for two reasons: (1). by continuity of $\hat f$ and $\varphi$, equation (4) decreases as $|\tau_{i+1}-\tau_{i}|$ decreases; (2). As shown in Theorem 1 and 2 of Song et al. 2023, smaller $\Delta\tau$ improves the approximation for the consistency loss in both consistency distillation and consistency training. Hence, $\epsilon$ can be smaller.

---

> > ### Comment · Reviewer_kTQM · 2025-04-04
> >
> > Thanks for your response. I still have concerns after reading the contents. Here are my questions.
> >
> > 1. I feel that Point 1 does not directly address my question. While I agree that the strategic design of the sampling schedule is important, the authors specifically claim there is a **trade-off** in selecting the **number** of sampling steps. This claim appears unrelated to the examples provided.
> >
> >     "trade-off means increasing the number of sampling steps does not necessarily lead to improved performance due to the influence of term (ii)". My main concern lies in this part. Take a trivial example. Let $f(x) = 2x + (-x)$. While the first part increases with $x$ and the other decreases, the total function is still strictly increasing.  Therefore, I find it too easy to claim a tradeoff in choosing the number of sampling steps directly from the theoretical result of Theorem 2.
> >
> > 2. I mention the case $\epsilon_{cm} \approx \tau$ because it's written in Line 339. If that's a failure case, why do you mention it there?
> >
> > 3. What do you mean by "fractional gains" in Rombach et al.?
> >
> > 4. Lastly, it's strange to compare two upper bounds with big $O$ notation and then conclude that one method is faster than the other. After all, these are only upper bounds, not tight estimates of actual performance. While this may not be a critical issue, the authors should be more careful in making such claims. Arguing that multi-step sampling is better than single-step sampling based solely on a comparison of upper bounds is problematic.
> >
> > ---
> > My concerns have been addressed after reading the newest rebuttal. Thus, I have adjusted my score accordingly. I do suggest the authors include a detailed discussion of the problems we have discussed in the revision, such as the accurate definition of the tradeoff, concrete examples, and comparison of rates. This will help the paper become more readable.

---

> > > ### Author Response · Authors · 2025-04-05
> > >
> > > We sincerely thank the reviewer for the opportunity to further discuss these points. Please find our responses to your comments below:
> > >
> > > 1. We appreciate the reviewer’s feedback and would like to clarify that the examples provided in our rebuttal are indeed related to our claim. Specifically, we included:
> > >    - two instantiations of our upper bound as theoretical examples;
> > >    - a summary of our simulation results in Appendix G;
> > >    - a summary of real-world experiments from Luo et al. (2023).
> > >
> > > All these examples support the observation that increasing the number of sampling steps in consistency models, without careful strategy, can degrade performance. We understand the reviewer’s concern and will ensure to use more precise language in a future revision.
> > >
> > > 2. To clarify, we respectfully note that **the case described in the review and the case described in our paper are not equivalent.** In our paper, the scenario where $\epsilon_{cm} \approx \tau$ does not necessarily indicate a failure case. In practice, the diameter $R$ of the data distribution may scale with the dimension. For example, natural images are supported on a hypercube $[0,1]^d$ where $d$ corresponds to the number of pixels and channels, and $R = \sqrt{d}$ in the worst case. In this context, an error bound of $O(1)$ is actually meaningful, as the trivial upper bound is $\sqrt{d}$, a potentially large quantity. On the other hand, the situation described in the review, where $\epsilon_{cm} \approx \Delta\tau$ and $R = O(1)$, does correspond to a failure case.
> > > 3. Regarding the comment that a constant reduction may not be significant, we respectfully disagree. In practice, even constant improvements in evaluation metrics such as the FID score can be quite impactful. For instance, Rombach et al. (LDM) demonstrated impressive results in image synthesis with FID improvements over prior works.
> > >    - On the CelebA-HQ benchmark (Table 1), LDM achieved an FID of 5.11, compared to baselines ranging from 7.16 to 15.8;
> > >    - in Table 3, LDM achieved 3.6 vs. 4.59–10.94;
> > >    - in Table 5, 2.4/4.3 vs. 5.2–15.2;
> > >    - in Table 7, 9.39 vs. 10.4–30.5.
> > >
> > > These examples show that constant-level improvements are indeed considered meaningful in the community.
> > >
> > > 4.
> > >    - **Regarding the big O notation:** we adopt big O notation primarily to simplify the presentation by hiding constants. Nonetheless, we believe the improvements to the upper bounds remain clear, even with the big O notation. In Case Study 1, we explicitly retain the constants in the leading terms and apply big O notation only to the lower-order terms. In Case Study 2, we highlight a reduction in rate, which clearly demonstrates an improvement despite the use of big O notation. Moreover, the exact bounds, including constants, can be readily derived by specifying the $t_j$’s in Theorem 2.
> > >    - We would also like to emphasize that our main contribution lies in establishing theoretical guarantees for consistency models while relaxing several strong assumptions made in previous works. We appreciate your feedback and will take greater care to ensure precise and clear writing in future revisions.

---

### Official Review · Reviewer_eZRV · 2025-03-17

**Overall Recommendation:** 3

**Summary:**

This paper analyzes consistency models—a recently introduced approach for accelerating sampling in diffusion-based generative models. Unlike classical diffusion models that rely on multiple iterative score-based updates, consistency models learn a direct mapping (“consistency function”) from noise to data while preserving the so-called self-consistency property. This enables both one-step sampling and optional multi-step refinement.

In this paper, the authors showed that if the self-consistency property holds approximately (quantified by a small “consistency loss”), then one-step or multi-step sampling from a CM can approximate the target data distribution in the Wasserstein-2 distance. Different from other relevant works, the analysis requires only mild assumptions on the data distribution (e.g., bounded support or sufficiently fast-decaying tails for the Wasserstein results, and log-smoothness for the TV results). The authors proved that two-step sampling can yield appreciable improvement over single-step sampling, but adding more than two steps leads to diminishing gains—mirroring empirical observations in prior work.

**Claims And Evidence:**

The theoretical claims follow from some widely used lemmas (e.g., chain rule of KL, Minkowski’s inequality for L2 errors) and do not appear to be overstated or overclaimed. There are no large leaps of logic, and each main theorem is accompanied by a clear proof outline.

Claim 1: Wasserstein Guarantees: The derivations rely on standard techniques—Pinsker’s inequality, data-processing inequalities, and mild assumptions (such as bounded or suitably light-tailed distributions). The authors' arguments are mathematically sound and are aligned with recognized results in diffusion modeling theory.
Claim 2: Total Variation Bounds: The authors show that smoothing with a small Gaussian convolution can produce a valid bound in TV distance. This theoretical technique is consistent with prior approaches in generative modeling when bridging the gap between pointwise errors and distributional overlap.

**Essential References Not Discussed:**

NA

**Ethical Review Concerns:**

There is no ethical review concerns since it is a theoretical work.

**Experimental Designs Or Analyses:**

This paper is a theory work and there is no experimental designs or analyses needed.

**Methods And Evaluation Criteria:**

This paper is a completely theoretical work and there is no benchmark or datasets involved or required.

**Other Comments Or Suggestions:**

Please refer to the "strength and weaknesses" section.

**Other Strengths And Weaknesses:**

1. Some proofs rely on dimension-free arguments (via Wasserstein and pinned distributions), but real-world data can be extremely high-dimensional. Whether the derived rates remain meaningful in large-scale real tasks is still an open question. I would like to ask the authors about the high-dimension issue, whether curse-of-dimension will make the derived rate in the high-dimensional case unreasonable in the real world scenario.

2. It would be very helpful if the authors can do some toy experiments to add some simple empirical ideas. Even a small synthetic or 2D test could give a sense of how the theoretical results manifest empirically. For example, the paper’s theory suggests that after two steps, gains diminish significantly. Could the authors share real-data experiments or references to confirm that this theoretical phenomenon fully matches practice?

3. You propose different strategies for picking sample times (e.g., geometric halving). Have you considered adaptive schedules dependent on learned \hat{f} or data statistics during sampling? Will it make any difference to the proof?

To sum up, this paper offers a valuable theoretical understanding of consistency models under mild data assumptions, particularly clarifying the trade-offs with multi-step sampling. The main theorems are carefully argued, the bounding steps are standard but precise, and the conclusion that “two-step sampling yields a notable boost while further steps add smaller improvements” is both practically and theoretically important.

**Questions For Authors:**

Please refer to the "strength and weaknesses" section.

**Relation To Broader Scientific Literature:**

Prior theoretical works (e.g., Lyu et al. 2023; Dou et al. 2024) examine consistency models but often under stronger Lipschitz assumptions or only variance-preserving SDEs. This paper’s novelty is in requiring fewer assumptions on the data distribution (“light tails” or “bounded support” rather than strict Lipschitzness) and analyzing multi-step sampling for general forward processes.

They also draw connections to analyses of probability flow ODE solutions in diffusion modeling (Chen et al. 2023, Li et al. 2024, etc.), but adapt these arguments to the direct one-step or few-step sampling approach.

**Theoretical Claims:**

Yes, I checked the correctness of the proofs.

---

> ### Author Rebuttal · Authors · 2025-04-01
>
> Thank you for your feedback. Please see our detailed responses below:
> 1. **Regarding high-dimension issue:** even when accounting for the implicit dependency on the dimension, our upper bound remains at most polynomial in dimension and thus does not suffer from the curse of dimensionality. For example, in Corollary 1, our result shows that given an $\epsilon$-accurate consistency function estimate, the Wasserstein distance $W_2(\hat P, P_{\text{data}})$ depends only on consistency error $\epsilon$ (assuming $\Delta \tau = 1$ for simplicity) and the diameter of the distribution $R$. Considering the implicit dependency on $d$:
>     - **the diameter $R$ grows at most polynomially with $d$:** if the ground-truth distribution $P_{\text{data}}$ has support on the $d$-dimensional hyper-cube $[0,1]^d$, then $R=\sqrt{d}$; if the ground truth distribution $P_{\text{data}}$ has support only on a low-dimensional manifold, e.g. $[0,1]\times \{0\} \times \cdots \times \{0\}$, then $R$ can be some constant.
>     - **the consistency error $\epsilon$ scales at most polynomially with $d$:** by definition, $\epsilon$ arises from a $d$-dimensional prediction problem and is expected to scale at most polynomially in $d$.
> Given these factors, we believe our bounds remain both reasonable and meaningful in high-dimensional real-world scenarios.
> 2. **Regarding experiments:** Please refer to our Appendix G for simulations related to Case study 1. We show that for two heuristic sampling strategies, increasing the number of sampling steps yields diminishing improvements or even degraded performance. In contrast, our sampling strategy nearly matches the best performance of the heuristic strategies with only two sampling steps. Furthermore, our theoretical upper bound closely approximates the Wasserstein distance observed in practice during sampling. Additionally, Tables 1 and 2 in Luo et al. (2023) show that in LCM, 2-step sampling significantly outperforms 1-step sampling in terms of FID score, while 2-, 4-, and 8-step sampling exhibit comparable performance. Notably, 8-step sampling even yields worse FID scores than 4-step sampling. These empirical findings further support our theoretical results.
> 3. **Regarding adaptive schedules:**
>     - In **Case Study 1**, we optimize the general upper bound in Theorem 2 by choosing $t_j$'s strategically and propose a two-step sampling strategy that adapts to both the problem parameters (the distribution diameter $R$) and the learned $\hat f$ (the consistency error $\epsilon_{\text{cm}}$);
>     - For **Case Study 2**, here is a strategical derivation for a sampling schedule. In order to adjust the $t_j$'s to minimize the error upper bound, we first estimate the lower bound of equation (12). For an arbitrary sampling strategy $t_1 \ge t_2 \ge \ldots \ge t_N$, the summation inside term (i) can be lower bounded by:
>   $$
>   \sum_{j=2}^N \frac{t_{j-1}^2}{t_j^2} \ge \sum_{j=2}^N \frac{t_j(t_{j-1}-t_j)}{t_j^2}
>   = \sum_{j=2}^N \frac{t_{j-1}-t_j}{t_j} \ge \int_{t_N}^{t_1} \frac{d t}{t} = \log \frac{t_1}{t_N}.
>   $$
>   A schedule defined by a geometric series approximates this lower bound well. Letting $t_j = t_N \rho^{N-j}$ with $\rho > 1$, we have $\sum_{j=2}^N \frac{t_{j-1}^2}{t_j^2} = \rho^2 \log_\rho\frac{t_1}{t_N} = \frac{\rho^2}{\log \rho}\log\frac{t_1}{t_N} \ge 2e\log\frac{t_1}{t_N}$, where equality holds when $\rho = \sqrt{e}$. Substituting this into equation (12), we obtain:
>   $$
>   2R \left(\frac{R^2}{4t_1^2} + \frac{2e\epsilon_{\text{cm}}^2}{4\Delta\tau^2}\left(\log t_1 + \log\frac{1}{t_N}\right)\right)^{1/4} + t_N\frac{\epsilon_{\text{cm}}}{\Delta\tau}.
>   $$
>   To minimize this expression, we choose $t_1 = \sqrt{\frac{1}{e}}\frac{R\Delta\tau}{\epsilon_{\text{cm}}}$.
>   As $t_N \to 0$, the second term decreases linearly, while the first term increases slowly. Therefore, a reasonable choice is $t_N = \Delta\tau$, a small constant.
>    - In addition, for any new sampling strategy, one only needs to specify the noise schedule $\{\alpha_t,\sigma_t^2\}$ and the sampling steps $(t_1, t_2, \ldots, t_N)$ in Theorem 2. The majority of the proof remains unchanged.

---

### Decision · Program_Chairs · 2025-05-01

**Decision:**

Accept (poster)

**Comment:**

This paper establishes convergence guarantees of consistency diffusion models. Compared to existing works, this study extends to variance exploding forward process and the convergence guarantees include both the Wasserstein and TV distances.

The results are theoretical and reviewers appreciated the soundness and correctness of the results, despite concerns on interpretation and presentation. Understanding consistency model is also recognized as an important direction, given existing results deviates from practical implementations.

Major concerns are 1) whether the assumptions are realistic and restrictive, 2) interpretation of the upper bounds, and 3) numerical validations. For the first concern, authors' response provides discussions on the dimension dependence of the consistency error. Viewing the problem as a prediction problem not necessarily leads to a polynomial dependence on dimension---for example, a nonparametric regression inherently gives rise to an exponential dependence, known as the curse of dimensionality. While this is relatively a minor point, the assumption that the consistency error is relatively small, mimics the setup in Lyu et al. In Dou et al., only a Lipschitz assumption on the consistency function is required. Several discussions are favorable. For the second concern, the generality of the theory is appreciated, therefore, it is possible that the upper bound is large if the consistency training failed, or discretization step size is not appropriate. Therefore, it is suggested to revise the discussion around the theory as promised by the authors. Third, Appendix G provides a toy numerical example aiming to validate the theory. As suggested by several reviewers, since the theory implies relationship between consistency loss $\epsilon_{cm}$ and $\Delta \tau$, authors may consider devise new consistency model implementation and demonstrate their performance. This can largely strengthen the paper's contribution.